# Alterations in Kidney Structures Caused by Age Vary According to Sex and Dehydration Condition

**DOI:** 10.3390/ijms232415672

**Published:** 2022-12-10

**Authors:** Susana Quirós Cognuck, Lucas Ferreira de Almeida, Wagner L. Reis, Márcia S. Silva, Gislaine Almeida-Pereira, Sandra V. Zorro, André S. Mecawi, Terezila Machado Coimbra, Lucila L. K. Elias, José Antunes-Rodrigues

**Affiliations:** 1Department of Physiology, Ribeirao Preto Medical School, University of Sao Paulo, Ribeirao Preto 14049-900, SP, Brazil; 2Department of Physiological Science, Center of Biological Sciences, Federal University of Santa Catarina, Florianópolis 88040-900, SC, Brazil; 3Medical Clinic Department, Ribeirao Preto Medical School, University of Sao Paulo, Ribeirao Preto 14048-900, SP, Brazil; 4Laboratory of Neuroendocrinology, Department of Biophysics, Escola Paulista de Medicina, Universidade Federal de Sao Paulo, Sao Paulo 04023-900, SP, Brazil

**Keywords:** aging, body composition, kidney, sex differences

## Abstract

Aging is a complex biological process, resulting in gradual and progressive decline in structure and function in many organ systems. Our objective is to determine if structural changes produced by aging vary with sex in a stressful situation such as dehydration. The expression of *Slc12a3* mRNA in the renal cortex, α-smooth muscle actin (α-SMA), and fibronectin was evaluated in male and female rats, aged 3 and 18 months, submitted and not submitted to water deprivation (WD) for 48 h, respectively. When comparing ages, 18-month-old males showed a lower expression of *Slc12a3* mRNA than 3-month-old males, and control and WD 18-month-old male and female rats exhibited a higher expression of α-SMA than the respective 3-month-old rats. Fibronectin was higher in both control and WD 18-month-old males than the respective 3-month-old males. In females, only the control 18-month-old rats showed higher fibronectin than the control 3-month-old rats. When we compared sex, control and WD 3-month-old female rats had a lower expression of *Slc12a3* mRNA than the respective males. The WD 18-month-old male rats presented a higher expression of fibronectin and α-SMA than the WD 18-month-old female rats. When we compared hydric conditions, the WD 18-month-old males displayed a lower relative expression of *Slc12a3* mRNA and higher α-SMA expression than the control 18-month-old males. Aging, sex, and dehydration lead to alterations in kidney structure.

## 1. Introduction

The kidneys are primarily organs that regulate the volume and composition of the internal fluid environment through the specialized sequence of functions in the different segments of the nephron. The number and length of the loops of Henle and the arrangement of the capillary circulation, as well as the feedback interactions between different parts of the nephron and the extrarenal control through nervous innervation and humoral agents, are important factors that operate throughout the entire kidney [1].

Aging is a complex biological process, with gradual and progressive decline in structure and function in many organ systems. In the kidney, morphological changes occur in the glomeruli, in the glomerular basement membrane, in the tubulointerstitium, and in renal vasculature with aging [2]. The structural changes in the kidney associated with hemodynamic changes include the loss of renal mass; the hyalinization of afferent arterioles; and the development of aglomerular arterioles, sclerotic glomeruli, and tubulointerstitial fibrosis [3].

Changes in the chemical composition of the glomerular basement membrane, related to increases in non-enzymatic glycosylation of proteins; increased expression of collagen, laminin, fibronectin, and thrombospondin; and changes in the degree of sulfation of glycosaminoglycans, are also observed [4].

In the injured kidney, interstitial infiltrate macrophages synthesize fibronectin and tubular cells change their phenotype to myofibroblasts, which are able to produce extracellular matrix components, such α-smooth muscle actin (α-SMA), a protein that is expressed normally in the renal cortex only by vascular smooth muscle cells [5].

In addition, renal aging is characterized by progressive tubular dysfunction, such as decreased sodium reabsorption, potassium excretion, and renal concentrating and diluting abilities [6]. The lowered ability to concentrate urine in aged rats may be related to the reduced level of sodium transporters [7]. However, among the sodium transporters, the thiazide-sensitive Na+/Cl- cotransporter (NCC), encoded by the *Slc12a3* gene, was not reduced by age in male rats, and the water deprivation condition resulted in either no increase or a reduced increase in the protein abundance of NCC [7,8,9].

Aging alters kidney structure, and older people are more at risk of dehydration because of the reduced thirst perception and the renal capacity to concentrate urine [10,11]. In addition, changes in hydromineral homeostasis in the elderly are also related to sex [11]. Moreover, the effects of estrogens on fluid regulation in older women are mediated in the kidney [12]. Our goal is to determine if structural changes produced by aging vary with sex in a stressful situation such as dehydration. We hypothesized that alterations produced in the kidney by aging show sexual differences and change with dehydration.

## 2. Results

### 2.1. Kidney and Body Weight

Table 1 provides the information about the left kidney and body weight of the animals. Both hydrated and dehydrated 3-month-old male and female rats had higher left kidney weights than 18-month-old male and female rats (Age: Male: F_1,32_ = 18.9, *p* < 0.001; Female: F_1,33_ = 63.6, *p* < 0.00001). Both dehydrated 3- and 18-month-old male rats presented higher kidney weights than hydrated rats (Hydration: F_1,32_ = 22.0, *p* < 0.0001). In both ages studied, the dehydrated male rats exhibited heavier left kidneys than the dehydrated female rats (Sex: 3-month-old: F_1,40_ = 8.7, *p* < 0.01; 18-month-old: F_1,25_ = 29.7, *p* < 0.0001).

Male and female rats at 18 months old showed heavier body weight than 3-month-old rats (Age: Male: F_1,31_ = 57.5, *p* < 0.0001; Female: F_1,31_ = 86.1, *p* < 0.0001). Dehydrated 3-month-old male and female rats had lighter body weights than hydrated 3-month-old male and female rats (Hydration: Male: F_1,31_ = 6.5, *p* < 0.05; Female: F_1,31_ = 13.6, *p* < 0.01). Both 3- and 18-month-old male rats presented higher body weights than 3- and 18-month-old female rats (Sex: 3-month-old: F_1,40_ = 250.2, *p* < 0.0001; 18-month-old: F_1,25_ = 37.0 *p* < 0.0001).

### 2.2. Sodium Plasma Concentration

Male and female rats submitted to WD showed a higher plasma sodium concentration than the respective control group (Hydration: Male: F_1,32_ = 12.6, *p* < 0.001; Female: F_1,34_ = 27.8, *p* < 0.00001). The 3-month-old male rats presented higher plasma sodium concentration than the respective females (Sex: 3-month-old: F_1,40_ = 7.8, *p* < 0.01).

### 2.3. The Relative Expression of Slc12a3 mRNA in the Renal Cortex

Both the control and WD 18-month-old males had a lower expression of *Slc12a3* mRNA than the 3-month-old males (Age: Control: t_10_ = 2.90, *p* < 0.05; WD: t_12_ = 3.12, *p* < 0.01; Figure 1A–D). The control and WD 3-month-old female rats both had a lower expression of *Slc12a3* mRNA than the respective males (Sex: Control: t_8_ = −4.10, *p* < 0.01; WD: t_10_ = −3.67, *p* < 0.01; Figure 1E–H). The water-deprived 18-month-old males presented a lower relative expression of *Slc12a3* mRNA than the hydrated old males (Hydration: t_24_ = 2.29, *p* < 0.05; Figure 1I–L).

### 2.4. Immunohistochemical Analysis

The immunohistochemical studies found a higher score for α-SMA in the cortical tubulointerstitium in 18-month-old animals than 3-month-old animals (Age: F_1,43_ = 111.05, *p* < 0.00001, Figure 2). WD 18-month-old males exhibited higher renal expression of α-SMA than control 18-month-old male rats (Hydric condition: F_1,43_ = 7.70, *p* < 0.01) and WD 18-month-old female rats (Sex: F_1,43_ = 5.74, *p* < 0.05, Figure 2).

Immunohistochemical analysis also revealed that the cortical tubulointerstitium expression of fibronectin was higher in control 18-month-old rats than control 3-month-old rats, as well as WD 18-month-old male rats than WD 3-month-old male rats (Age: F_1,43_ = 72.77, *p* < 0.0001, Figure 3). WD 18-month-old male rats presented a higher expression of fibronectin than WD 18-month-old female rats (Sex: F_1,43_ = 13.18, *p* < 0.001, Figure 3).

## 3. Discussion

As is already known and we confirm in our results, renal mass declines with aging, and sex also influences kidney weight [13,14]. The increase in renal weight observed in dehydrated 3-month-old animals and 18-month-old males confirms the results reported by Bankir et al., who indicated that weight gain is influenced by the urine concentration mechanism and that the outer medulla suffered changes such as increased volume of the epithelium in the thick ascending limb of Henle’s loops and collecting ducts [15]. Despite this, we found that 18-month-old females did not show changes in kidney weight with dehydration, but more research is required to understand the mechanism that explain this interesting finding.

The 3-month-old female rats presented a lower relative expression of *Slc12a3* mRNA in the renal cortex than 3-month-old males, as expected, since estradiol decreases the expression of NCC in the membrane of the distal convoluted tubule [16]. The absence of sexual difference in the relative expression of *Slc12a3* mRNA in the renal cortex observed in 18-month-old animals may be because the female rats were in reproductive senescence.

The expression of NCC decreased in the renal cortex of 18-month-old male Wistar rats, possibly because testosterone is decreased in these animals as already shown by our group elsewhere [17].

This hormone increases the plasma levels of fibroblast growth factor-23, which increases the expression of NCC in the kidney [17,18,19].

Interestingly, water deprivation decreased the expression of NCC in the renal cortex of dehydrated 18-month-old males, but it did not change the expression of the NCC of 3-month-old rats or 18-month-old females. These results may reflect a balance between gonadal hormones and arginine vasopressin, which increase the plasma level during dehydration and NCC expression [17,20]. Therefore, in 18-month-old male rats, the decrease in testosterone possibly exerts a greater effect on the expression of NCC in relation to the increase in plasma vasopressin.

According to our immunohistochemical results, aging is characterized by renal structural changes, such as alterations in the extracellular matrix because of increased fibronectin and α-SMA [21,22]. Interestingly, and described for the first time, the water-deprived 18-month-old male rats showed higher renal expression of α-SMA than control 18-month-old male rats. Similar results were observed in the lizard *Uromastyx acanthinura*, with the expression of α-SMA surrounding the collecting duct, indicating that this renal structural characteristic is involved in body water economy and may be considered as an adaptive mechanism to resist dehydration in an arid environment [23].

The sexual dimorphism observed in the immunohistochemical labeling of α-SMA and fibronectin in the animals submitted to dehydration could be explained by the fact that estrogen stimulates Angiotensin II receptor types, and this receptor is associated with increased nitric oxide production, whose chronic inhibition is related to elevated expressions of fibronectin and α-SMA [24,25,26,27,28].

In conclusion, alterations in the kidney structure confirm the effect of accumulation of injury with aging. In addition, water deprivation alters the kidney structure of 18-month-old male rats. Furthermore, altered renal structures in 18-month-old animals showed sexual dimorphism when the animals were subjected to water deprivation. However, sexual differences in the relative *Slc12a3* mRNA expression observed in 3-month-old rats disappear in 18-month-old rats.

## 4. Materials and Methods

### 4.1. Animal Model and Experimental Design

Female and male Wistar rats were acquired from the animal facility located at the University of Sao Paulo, Ribeirao Preto Campus, SP, Brazil. The animals were kept under a controlled temperature of 23 ± 2 °C and exposed to a 12:12 h light–dark cycle (light on: 6 a.m. to 6 p.m.) with tap water and standard pelleted food (QuimtiaNuvilab^®^, Colombo, Brazil) ad libitum. Experimental methods were performed in the morning (8:00–11:00 h). All procedures were approved by the Ethics Committee for Animal Use of the School of Medicine of Ribeirao Preto, University of Sao Paulo (protocol # 014/2014-1) and conducted according to the “Guide for the Care and Use of Laboratory Animals” (NIH; Publication No. 85-23, revised 1996).

The lifespan of the Wistar rat colony in our animal facility is ~2 years, which correlated with the human life expectancy of ~80 years. Therefore, we infer that the age of 18 months corresponds to the sixth decade of human life. At 18 months, the female rats are in permanent diestrus.

### 4.2. Experimental Methods

Male and female Wistar rats at the age of 3 and 18 months were submitted or not submitted to WD for 48 h. They had free access to food. After that, the animals were weighed and euthanized by decapitation, by an experienced technician, quickly and with care not to cause stress. Blood from the trunk was collected in chilled tubes containing heparin (10 μL/mL of blood) to measure plasma sodium.

### 4.3. Determination of Plasma Sodium

An electrolyte analyzer (9180 Electrolyte Analyzer, Roche, IN, USA) was used to determine the plasma sodium concentration.

### 4.4. Tissue Collection

After euthanasia, both kidneys were collected and cleaned of connective tissue. The left kidneys were weighted and fixed for immunohistochemical analysis, and the right kidneys were used for the relative gene expression of mRNA.

### 4.5. Microdissection, RNA Isolation, and Semiquantitative Real-Time PCR

After the right kidney was dissected with a sagittal cut with the aid of a sterile microdissection needle (5 mm internal diameter), a sample of the renal cortex was dissected and placed in sterile tubes and stored at −70 °C.

Total RNA was extracted using the RNeasy Mini Kit-Qiagen^®^ and treated with DNase using the DNA-free TM kit (Ambion^®^, now Life Technologies, CA, USA). The RNA purity and concentration were verified in a spectrophotometer (SpectraMax^®^ i3x Multi-Mode Microplate Reader).

The cDNA synthesis was made from 250 ng of RNA using the High-Capacity cDNA Reverse Transcription kit (Applied Biosystems^®^ Curitiba, Brazil). Reverse transcription was made in a thermocycler (GeneAmp PCR System 9600, Applied Biosystems, Curitiba, Brazil) at 25 °C for 10 min and at 37 °C for 120 min. After that, the samples were stored at −20 °C.

The QT-PCR was performed in the 7500 QT-PCR System (Applied Biosystems^®^) using Taqman^®^ assays (Applied Biosystems^®^, Curitiba, Brazil). Each sample was run in triplicate. *Rat ACTB* (actin, beta; Rn00667869_m1) was the internal control gene and *Slc12a3* (Rn01531762_m1) was the target gene. The relative expression of the target gene was determined based on the threshold cycle (Ct). The results were analyzed using the ΔΔCt method.

### 4.6. Immunohistochemical Studies

Kidneys were fixed for 24 h in methacarn solution (60% *v/v* methanol, 30% *v/v* chloroform, and 10% *v/v* acetic acid) followed by washes in 70% *v/v* alcohol and embedding in paraffin; sectioned into 4 μm slices; deparaffinized; and incubated overnight at 4 °C with the following antibodies: 1/1000 anti-α-SMA (Dako Corporation, Glostrup, Denmark) or 1/500 anti-rat fibronectin (Chemicon International Inc., Temecula, CA, USA). The reaction product was detected with an avidin–biotin–peroxidase complex (Vector Laboratories, Burlingame, CA, USA). The color reaction was developed with 3,3-diamino-benzidine (Sigma Chemical Company, St. Louis, MO, USA). Counterstaining was performed with methyl green, and after that, the material was dehydrated, and mounted.

For the evaluation of the immunoperoxidase staining for α-SMA and fibronectin, the cortical 30-grid field (measuring 0.100 mm^2^ each) was semiquantitatively graded, and the mean score per kidney was calculated. The scores mainly reflect changes in the extent rather than in the intensity of staining and depend on the percentage of the grid field showing positive staining: 0 = absent or <5% staining; 1, 5–25%; 2, 25–50%; 3, 50–75%; and 4, >75% staining [13].

The analyses in the cortical tubulointerstitium were identified at 40x magnification (Zeiss microscope supplied with a DC 200 Leica digital camera affixed to a contrast enhancement device) using a double-blind method. Counts were performed randomly throughout all fields.

### 4.7. Statistical Analyses

Results are presented as means (Standard Deviation (SD)), and for immunolocalization, as median and interquartile range (25–75th). Data were plotted using GraphPad Prism version 7 (GraphPad Software, CA, USA) and analyzed using Statistica (StatSoft, USA) and SPSS software (IBM, USA). The Shapiro–Wilkʹs W test was carried out to confirm the assumption of the normality of the distribution. The ROUT test, with Q = 10%, was used to identify and remove outliers in the data of the relative expression of *Slc12a3* mRNA in the renal cortex. The statistical significance of the difference between the means of the studied groups was assessed by the *t*-Student test for independent groups for the relative expression of *Slc12a3* mRNA in the renal cortex; and for the immunohistochemical analysis, sodium plasma concentration, and kidney and body weight, by the analysis of variance (ANOVA) factorial followed by the Newman–Keuls or Duncan or Games-Howell post-test, where appropriate. The independent variables were age, sex, and water deprivation condition. The significance level of *p* < 0.05 (two-tailed) was adopted.

### 4.8. Study Limitations

Hormone concentration of testosterone was assessed and reported previously [18], but estradiol plasma concentration was not evaluated.

## Figures and Tables

**Figure 1 ijms-23-15672-f001:**
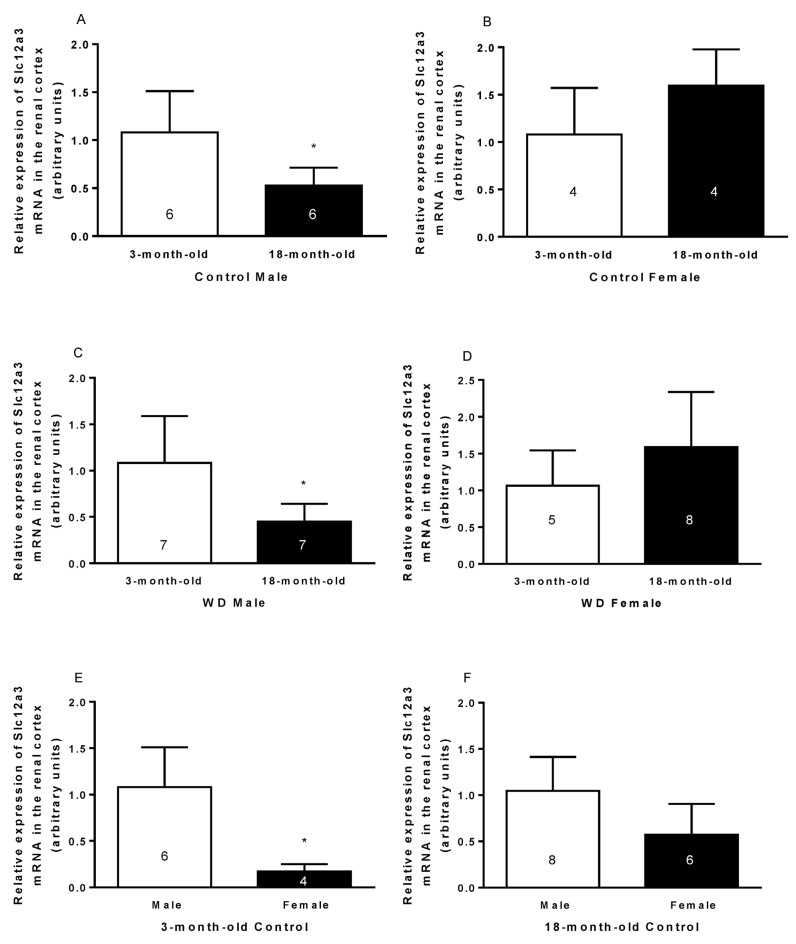
Relative *Slc12a3* mRNA expression in the renal cortex of 3- and 18-month-old control and water-deprived (WD) male and female rats. Comparing by age in the control male (**A**) and female rats (**B**) and the WD male (**C**) and WD female rats (**D**). Comparing by sex in the 3-month-old (**E**) and 18-month-old control rats (**F**) and 3-month-old (**G**) and 18-month-old WD rats (**H**). Comparing by hydric condition in the 3-month-old (**I**) and 18-month-old male rats (**J**) and 3-month-old (**K**) and 18-month-old female rats (**L**). Data are presented as means (SD), * *p* < 0.05 relative to the reference group. Student’s unpaired t-test. Number (n) of the sample is indicated inside the column.

**Figure 2 ijms-23-15672-f002:**
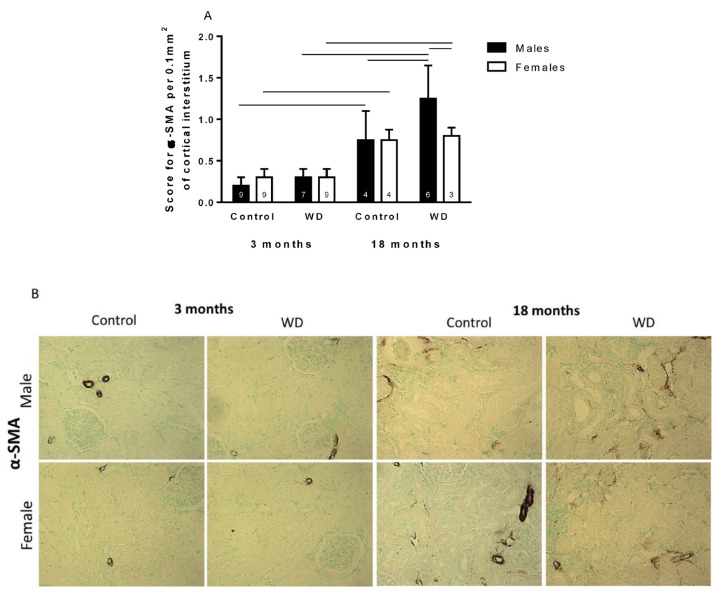
Score (**A**) and representative images (**B**) for α-SMA per 0.1 mm^2^ of cortical interstitium in control and water-deprived (WD) 3- and 18-month-old male and female rats (magnification 400×). Data are expressed as median and interquartile range (25–75th) ‘_____’: *p* < 0.05 among the indicated groups. Factorial ANOVA followed by Newman–Keuls post-test. Number (n) of animals is indicated inside the column.

**Figure 3 ijms-23-15672-f003:**
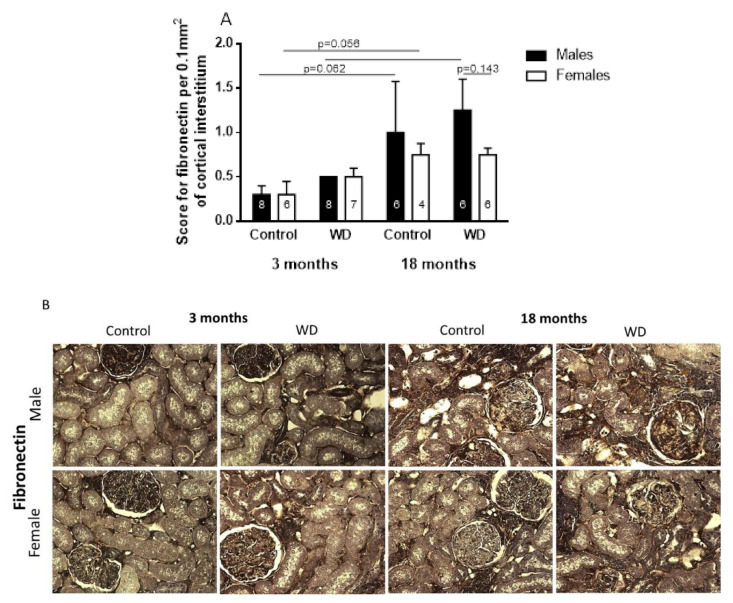
Score (**A**) and representative images (**B**) for fibronectin per 0.1 mm^2^ of cortical interstitium in control and water-deprived (WD) 3- and 18-month-old male and female rats (magnification 400×). Data are expressed as median and interquartile range (25–75th) ‘_____’: *p* < 0.05 among the indicated groups. Factorial ANOVA followed by Duncan post-test. Number (n) of animals is indicated inside the column.

**Table 1 ijms-23-15672-t001:** Left kidney and body weight of normal (Ctrl) and dehydrated (WD) 3- and 18-month-old male (M) and female (F) rats.

Parameters	Sex	Age (Months)			
		3		18	
		Ctrl	WD	Ctrl	WD
Number of Animals	M	10	10	8	8
	F	12	12	7	6
Kidney weight (g/100g bw)	M	0.35 (0.02) ^ac^	0.38 (0.03)^bcg^	0.31 (0.02) ^ad^	0.35 (0.02)^bdh^
	F	0.33 (0.02) ^e^	0.35 (0.02) ^fg^	0.29 (0.03) ^e^	0.28 (0.02) ^fh^
Body weight (g)	M	561.5 (39.5) ^iot^	476 (49.4) ^kom^	810.1 (147.9)^iv^	733 (123) ^ky^
	F	383.7 (19) ^pjt^	334.2 (20.3)^qjm^	550 (98.6) ^pv^	466.8 (59.4) ^qy^
Na^+^ _Plasma_ (mEq/L)	M	131.3 (10.9) ^xz^	143.5 (4.7) ^x^	138.6 (11.8)	146.6 (3.2)
	F	138.0 (3.1) ^rz^	147.6 (4.6) ^r^	141.6 (2.1) ^s^	145.7 (4.7) ^s^

Values are expressed as means (SD). Averages with the same superscript letters differ significantly *p* < 0.05. ANOVA factorial followed by the Newman–Keuls post-test or Games-Howell.

## Data Availability

Not applicable.

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
