# Peer review of "Alterations in Kidney Structures Caused by Age Vary According to Sex and Dehydration Condition"

_ijms, 2022, doi:10.3390/ijms232415672_

Round 1

Reviewer 1 Report

This is nice work investigating the influence of age, gender, and dehydration on markers of renal rat structure. The paper is correctly written, and conclusions are based on results. However, several concerns arose:

Minor comments:

- considering the main role of sex hormones in explaining differences within and between sexes is probable but is rather speculative because hormone concentrations were not assessed. This should be underlined in the lacking section of study limitations

 - please specify why the number of rats changed during the study.

- the number of 18-month-old animals is very low (<10) in each specified group. In addition, the ROUT test could delete up to 10% of data - maybe it should not be used? In this condition, a sample size calculation with the power of the ANOVA test should be presented.

- dependent variables were compared. Was the appropriate test used?

Author Response

Answer to reviewers

Manuscript ID: ijms-1967048

Type of manuscript: Brief Report

Title: Alterations in kidney structures caused by age vary according to sex
and dehydration condition

Corresponding Author: Susana Quirós Cognuck, Lucas Ferreira de Almeida

Authors: Susana Quirós Cognuck, Lucas Ferreira de Almeida, Wagner L
Reis, Márcia S Silva, Gislaine Almeida Pereira, Sandra v Zorro, André S
Mecawi, Terezila M. Coimbra, Lucila L K Elias, José Antunes Antunes Rodrigues

We would like to thank the referees for their comments that help to improve our manuscript.

Reviewer #1:

  1. Considering the main role of sex hormones in explaining differences within and between sexes is probable but is rather speculative because hormone concentrations were not assessed. This should be underlined in the lacking section of study limitations.

Answer: In the revised version, we included the clarification that the hormone concentration of testosterone was assessed and reported previously in Quirós Cognuck S, Reis WL, Silva MS, et al (2020) Sex- and age-dependent differences in the hormone and drinking responses to water deprivation. Am J Physiol - Regul Integr Comp Physiol 318:R567–R578. https://doi.org/10.1152/AJPREGU.00303.2019, but estradiol plasma concentration was not evaluated. (Line 247-249).

  1. Please specify why the number of rats changed during the study.

Answer: The number of rats changed during the study because during the develop of methodology some samples were lost.

  1. The number of 18-month-old animals is very low (<10) in each specified group. In addition, the ROUT test could delete up to 10% of data - maybe it should not be used? In this condition, a sample size calculation with the power of the ANOVA test should be presented.

Answer: In the revised version was included the suggestion of not used the ROUT in the case of data of ANOVA test (Line 117-118, 239 and figure 3A)

  1. Dependent variables were compared. Was the appropriate test used?

Answer: Yes, the test used were selected according to nature of the data, and each case we made test to select the best test to use according to the data, for example, test de normality, variance homogeneity, etc. Also, the test selected were chosen considering the number of dependent and independent variables.

Reviewer 2 Report

Overall, this is interesting study, however, the investigators need to address the term "dehydration"; given that this term means the loss or removal of water or reduction in the amount of water in the body; but in the methods, the investigators did not describe how the serum sodium of the animal models, no confirmation of hypernatremia and how they confirmed the dehydration state? Is it dehydration or volume depletion, needs to be additionally clarify and discuss.

Author Response

Answer to reviewers

Manuscript ID: ijms-1967048

Type of manuscript: Brief Report

Title: Alterations in kidney structures caused by age vary according to sex
and dehydration condition

Corresponding Author: Susana Quirós Cognuck, Lucas Ferreira de Almeida

Authors: Susana Quirós Cognuck, Lucas Ferreira de Almeida, Wagner L
Reis, Márcia S Silva, Gislaine Almeida Pereira, Sandra v Zorro, André S
Mecawi, Terezila M. Coimbra, Lucila L K Elias, José Antunes Antunes Rodrigues

We would like to thank the referees for their comments that help to improve our manuscript.

Reviewer #2:

  1. Overall, this is interesting study, however, the investigators need to address the term "dehydration"; given that this term means the loss or removal of water or reduction in the amount of water in the body; but in the methods, the investigators did not describe how the serum sodium of the animal models, no confirmation of hypernatremia and how they confirmed the dehydration state? Is it dehydration or volume depletion, needs to be additionally clarify and discuss.

Answer: In the revised version was included the suggestion (Line 101-105, 185-190, 248, and Table 1).
